# Occluded 3D Object Reconstruction via Masked Multi-view Volumetric Transformer

## Abstract

Recent advancements in image-to-3D reconstruction and generation have yielded remarkable progress. However, applying these methods to occluded objects in cluttered scenes remains challenging due to the incomplete information in occluded areas. To tackle this issue, we present a feed-forward method for reconstructing 3D occluded shapes using a data-driven approach. Our method utilizes a large-scale dataset of cluttered scenes and incorporates multi-view occlusion-aware 3D reconstruction through a Transformer architecture that draws inspiration from masked autoencoders. Our model, *Masked Multi-view Volumetric Transformer*, utilizes global reasoning from arbitrary number of multi-view 2D image information and cross-attention between 3D-lifted obstacle mask volumes and volumetric latents, enabling the model to predict information for occluded regions accurately. Furthermore, we have created a synthetic cluttered scene dataset comprising ∼30,000 scenes with Objaverse objects, designed to illustrate various occlusion scenarios. Our approach surpasses previous methods in predicting complete shapes from occluded images of unseen objects, achieving completed mesh extraction in five seconds.

## 1 Introduction

Reconstructing 3D shapes and synthesizing novel views from multi-view images are critical tasks in fields such as 3D computer vision (Hong et al., 2023; Tochilkin et al., 2024; Xu et al., 2024b;a; Chen et al., 2024a) and autonomous driving (Li et al., 2023). Recently, approaches utilizing the Transformer (Vaswani, 2017) architecture have successfully produced high-quality 3D reconstructions and novel view images with short runtimes. In particular, 3D reconstruction methods from single image (Hong et al., 2023; Tochilkin et al., 2024) and multiple images (Xu et al., 2024b;a; Chen et al., 2024a) have achieved high-quality 3D reconstruction or novel view synthesis in seconds. These methods typically extract latent image features using a pre-trained image encoder and employ attention mechanisms to establish relationships between image features and implicit 3D representations, such as triplane representations or Gaussian volumes.

Nonetheless, reconstructing occluded areas remains a significant challenge for these methods. While they excel in single-object settings, accurately predicting the entire shape of each object in cluttered scenes remains challenging due to occlusions between objects. Several approaches (Chen et al., 2024b; Weber et al., 2024) have sought to address this issue by inpainting the occluded areas of the target object using image generation models such as Stable Diffusion (Rombach et al., 2022). Despite the strong performance of image generation models, large inpainting masks can generate unintended background artifacts, resulting in incorrect 3D shape predictions. This challenge also restricts inpainting methods to using human-guided, tight masks around targets, which is impractical. Moreover, multi-view image inpainting often yields inconsistent generations across different viewpoints, degrading the quality of reconstruction and view synthesis results. For these reasons, using inpainting-based methods for general occlusion reconstruction remains impractical.

From the restrictions of inpainting-based methods, Amodal3R (Wu et al., 2025) suggests a new pipeline based on a 3D object generation model (Xiang et al., 2025). Using the mask-weighted cross-attention in the DiT (Peebles & Xie, 2023) structure, the model can generate better completed 3D shapes than methods that utilize 2D image completion. Nevertheless, Amodal3R is also susceptible

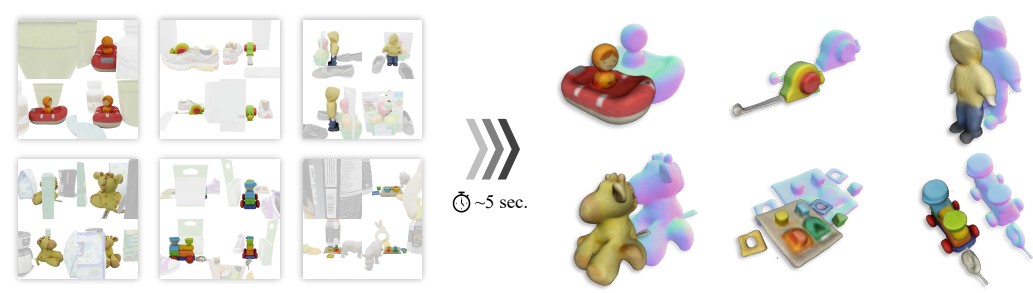

**Occlusion** from Cluttered Scene      ⏱ ~5 sec.      **Feed-forward** 3D Shape Reconstruction

Figure 1: We present a feed-forward 3D reconstruction method for occluded objects. Our approach ensures multi-view consistency while recovering the missing image information due to occlusion. As a result, our model performs highly at zero-shot 3D occluded shape reconstruction in under five seconds. To improve visibility, we intentionally brighten non-target regions in the left images.

to generative artifacts, which can result in outputs that deviate significantly from the input image conditions.

In this paper, we propose a faster and more generalizable method for 3D amodal completion based on the multi-view Transformer architecture, *Masked Multi-view Volumetric Transformer (MMVT)*, which addresses current challenges and enhances the practicality of reconstructing occluded areas. Our approach consists of two key components: a data-driven approach that utilizes our large-scale cluttered scene dataset, and a multi-view consistent occlusion-aware image-to-3D reconstruction pipeline.

Inspired by recent developments in masked autoencoders within the image domain (He et al., 2022; Weinzaepfel et al., 2022; 2023), we propose a multi-view consistent occluded object reconstruction architecture based on the MAE-like latent information reconstruction method. Achieving multi-view consistency is essential for accurate 3D shape reconstruction and novel view synthesis from multi-view images. We introduce a multi-view MAE-like latent reconstruction method that employs alternating attention structure, inspired by VGGT (Wang et al., 2025), which includes full self-attention over concatenated multi-view tokens and frame-level self-attention layers. Global reasoning with self-attention across all image features, modulated with Plücker rays, allows the prediction of missing latent information in occluded regions while ensuring multi-view consistency.

We also propose an occlusion-aware feed-forward reconstruction model, which enables 3D-aware handling of occlusions. For each viewpoint, corresponding 3D-lifted obstacle masks could work as an indicator of the occluded voxels in the 3D space. With the cross-attention between the mask volumes and the volumetric latent, our model enables accurate occlusion-aware 3D reconstruction.

Moreover, a large and diverse dataset of cluttered scenes is essential for learning general object shapes and understanding the complexities of occlusion in a data-driven approach. However, existing datasets featuring real-world scenes (Xiang et al., 2018; Kaskman et al., 2019; Tyree et al., 2022) consist of only a limited number of objects, which is insufficient for training generalizable shape and occlusion representations across a wide range of objects. To address this limitation, we propose a synthetic dataset comprising 29,358 scenes featuring various objects from Objaverse (Deitke et al., 2023), which enables our model to generalize effectively in reconstructing occluded areas.

In a zero-shot evaluation using our evaluation dataset consisting of unseen objects from Google Scanned Objects (GSO) (Downs et al., 2022) dataset, we observed that our method could reconstruct occluded regions even when faced with larger occlusions than those manageable by previous inpainting-based methods. Our model is highly effective, as it can produce the whole shape of hidden objects in just 5 seconds, making it significantly quicker than earlier approaches.

We summarize our contributions as follows:

- We propose a *Masked Multi-view Volumetric Transformer* architecture for multi-view consistent token reconstruction, utilizing global reasoning through self-attention over concatenated multi-view tokens and volumetric amodal completion via cross-attention with 3D obstacle masks.

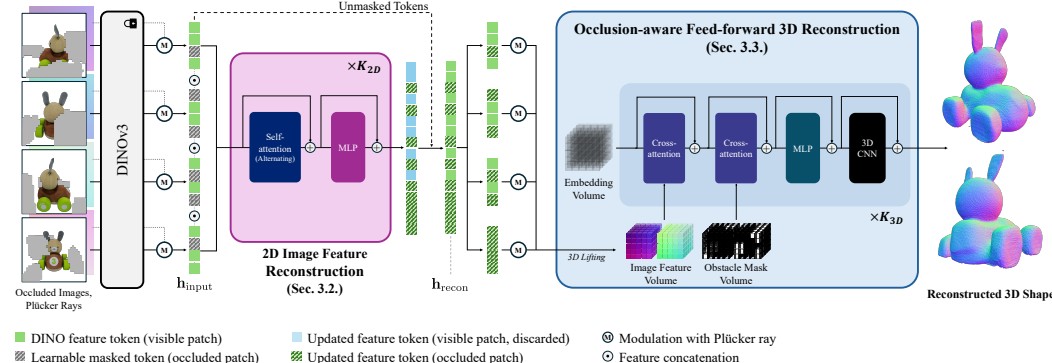

Figure 2: **Our proposed *Masked Multi-view Volumetric Transformer (MMVT)* method.** Our MAE-like method reconstructs missing latent representations for image feature tokens at masked patches of the occluded region by reasoning from multi-view information. Additionally, the 3D lifted obstacle mask volume guides the model to identify potentially occluded voxels in 3D space.

- We constructed a large-scale synthetic dataset of cluttered scenes for training and evaluation using various objects from Objaverse (Deitke et al., 2023) and GSO (Downs et al., 2022), enabling a data-driven approach for understanding complex occlusions.
- Our method effectively reconstructs occluded regions in unseen datasets, requiring only five seconds to produce completed 3D shapes for target objects.

## 2 RELATED WORK

**Large reconstruction models.** The advent of large-scale 3D datasets (Deitke et al., 2023; 2024) has empowered learning-based models to perform 3D reconstructions from single or few-view images. Large reconstruction models (LRMs) (Hong et al., 2023; Tochilkin et al., 2024) leverage a scalable transformer architecture to map a single image to an implicit 3D triplane NeRF, effectively learning generic 3D priors. Building on the capabilities of diffusion models, several approaches (Xu et al., 2024a; Li et al., 2024) have used multi-view diffusion techniques to synthesize additional views, extending LRM into a sparse-view reconstruction framework. Recently, LGM (Tang et al., 2025) and GRM (Xu et al., 2024b) have improved rendering efficiency by adopting 3D Gaussian representations with transformer or U-Net architectures instead of the triplane representation. Similarly, LaRa (Chen et al., 2024a) utilizes 2D Gaussian representations, enhancing mesh reconstruction quality compared to 3D Gaussian splatting while maintaining efficient rendering.

Despite these advancements, most feed-forward models are mainly designed for single-object reconstructions. This limits their ability to reconstruct occluded objects in more complex scenes, highlighting the need for effective occlusion handling methods in image-to-3D reconstruction.

**Occlusion-aware reconstruction methods.** Recent studies aim to enhance reconstruction models for handling occluded objects. A common strategy for handling occluded objects involves a two-stage pipeline (Chen et al., 2024b; Han et al., 2024; Dogaru et al., 2024; Ozguroglu et al., 2024; Hu et al., 2024): first, completing the missing regions in 2D using 2D diffusion priors, and then utilize off-the-shelf image-to-3D models from these completed images. However, this pipeline's critical challenge is multi-view inconsistency, as independent 2D completions can introduce significant artifacts to the final 3D shape. Various methods attempt to solve this consistency issue within the two-stage framework. NeRFiller (Weber et al., 2024) uses a grid prior to inpaint multiple views simultaneously, while ObjFiller-3D (Feng et al., 2025) treats views as a video sequence to enforce temporal coherence. Despite these improvements, they remain vulnerable to error propagation from the 2D stage. Departing from this 2D-centric pipeline, other works perform completion directly in 3D. Amodal3R (Wu et al., 2025) adapts a 3D generative diffusion model to complete shapes in a latent space, which ensures 3D consistency but suffers from the slow, iterative nature of diffusion models. The method by Cho et al. (2025) offers a fast, regression-based alternative that jointly segments and reconstructs the shape from a single image, though its architecture is not designed to fully leverage multi-view inputs.

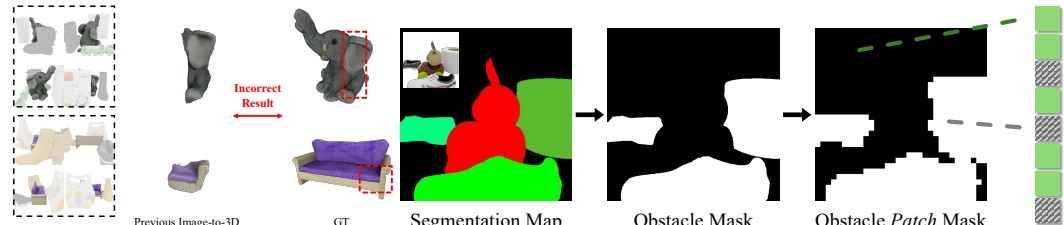

Previous Image-to-3D     GT     Segmentation Map     Obstacle Mask     Obstacle *Patch* Mask

(a) Examples of incorrect reconstruction cases for occluded images from the previous method (Chen et al., 2024a).

(b) Generation of obstacle mask from given instance segmentation map, for our *Masked Multi-view Volumetric Transformer*. Obstacle masks are patchified with the same patch size with image features.

Figure 3: **Overall illustration of the example of incorrect results and our masking strategy.**

Our work addresses these gaps, proposing a fast, feed-forward model specifically designed for multi-view inputs, which reconstructs complete 3D shapes end-to-end without relying on separate 2D inpainting modules or time-consuming iterative generation.

**MAE-like architectures for 3D reconstruction.** Recent advances that train ViT (Dosovitskiy et al., 2021) with a masked autoencoder (He et al., 2022) training scheme have enabled ViTs to learn robust image representations by reconstructing masked tokens. Building on this, VoxFormer (Li et al., 2023) employs an MAE-like architecture to jointly predict occupancy and semantic segmentation for occluded regions, resulting in improved performance compared to previous methods. Inspired by VoxFormer, OctMAE (Iwase et al., 2024) integrates an octree framework with an MAE-based design to reconstruct scenes involving multiple occluded objects from an RGB-D image. This method achieves efficiency through sparse 3D operations guided by depth information.

Our work aims to generate complete objects within seconds from multi-view images of occluded objects, even when occlusion obscures a substantial portion of the object in multi-view inputs.

## 3 METHOD

### 3.1 PROBLEM SETUP

Our pipeline aims to predict a Gaussian volume $\mathbf{V}_{\mathcal{G}}$ from occluded images that represent the complete shape of an occluded object. Given $N$ sparse-view images $\mathbf{I} = (I_1, \cdots, I_N)$ from the cluttered scene, along with their corresponding instance segmentation masks $\mathbf{M} = (M_1, \cdots, M_N)$ and camera parameters $\pi = (\pi_1, \cdots, \pi_N)$ for each image, the output of our model is a Gaussian volume $\mathbf{V}_{\mathcal{G}}$ that represents the complete shape of the target object.

To generalize our problem settings more practically, we use an obstacle object mask as our masking strategy. The obstacle object mask is the union of instance masks except for the target object mask, denoted as $\mathbf{M}_{\mathrm{obs}} = \bigcup M$, where $\mathbf{M}_{\mathrm{obs}}$ refers to the obstacle mask and $M \in \mathbf{M} \setminus \{M_{\mathrm{target}}\}$. The mask represents the occluders' region in the corresponding image for our occlusion reconstruction pipeline, which may include the area of the occluded parts of the target object. The mask is then patchified as shown in fig. 3b, indicating patch positions for determining where to reconstruct.

Unlike previous methods that rely on human-guided masks, our approach utilizes larger, automatically generated masks that require no human intervention. This enables a more generalizable reconstruction of the occluded area.

### 3.2 RESOLVING OCCLUSIONS WITH MASK TOKENS

Due to missing information from occluded regions, existing 3D reconstruction methods (Xu et al., 2024a; Chen et al., 2024a) cannot accurately predict the shapes of these regions, as illustrated in fig. 3a. The key challenge in recovering the occluded parts is: *How to predict the image information of the occluded regions*? Several feed-forward image-to-3D reconstruction methods (Hong et al., 2023; Li et al., 2024; Xu et al., 2024a; Tochilkin et al., 2024; Chen et al., 2024a) utilize pretrained image encoder (Caron et al., 2021) features to obtain detailed structural and texture information. However, we found that while image features provide appropriate feature tokens for the visible parts

of objects, the non-existence of feature tokens from the occluded regions negatively impacts object shape prediction. This leads to incorrect shape reconstructions, as shown in fig. 3a.

To address this problem, we propose a multi-view MAE-like architecture that reconstructs the absent latent from visible feature tokens in a data-driven manner, as shown in the left part of fig. 2. We employ a multi-view MAE structure, comprised of global and frame-level self-attention over concatenated multi-view tokens, to achieve the multi-view consistency required for accurate 3D reconstruction from multi-view images. This structure enables consistent latent completion across multiple images.

Given RGB images, we apply a pre-trained DINOv3 (Siméoni et al., 2025) image encoder to extract per-view image features $\mathbf{h}$, following Hong et al. (2023); Xu et al. (2024a), and inject Plücker ray directions via adaptive layer normalization. After Plücker ray modulation, we obtain patch-wise feature tokens of the $j$-th image, denoted as $\mathbf{h}_j \in \mathbb{R}^{L \times d_E}$, where $L$ is the number of patches and $d_E$ is the dimension of the latent features. This modulation allows for considering camera poses to reconstruct multi-view feature tokens.

Multi-view consistency in token reconstruction is crucial for accurately predicting the complete shape of an object when reconstructing an occluded object from multi-view images. As shown in fig. 3a, inconsistent reconstructions across views can result in artifacts or incorrect shapes. CroCo (Weinzaepfel et al., 2022) proposed an MAE method for cross-view completion by employing decoders with alternating self-attention and cross-attention to integrate stereo-view information (CrossBlock) or by using self-attention with concatenating two input sets from stereo images (Cat-Block). Inspired by the CatBlock-based decoder, we developed a multi-view MAE-like feature token reconstruction method utilizing Transformer blocks with full self-attention layers.

In contrast to CroCo, where occluded areas are present in only specific images, our scenario allows for occluded areas in every image. Thus, tokens from all images require cross-view reasoning across all pairs of images. This is done by global self-attention layers in our multi-view Transformer. We concatenate the modulated image features to complete the learnable mask tokens and apply full self-attention to this cross-view global reasoning. Given masked image feature tokens $\mathbf{h}_{\text{masked}}$, we concatenate the feature tokens for cross-view reasoning as $\mathbf{h}_{\text{input}} = \text{Concat}(\mathbf{h}_1, \ldots, \mathbf{h}_N)$. The concatenated tokens are fed into $K$ Transformer blocks, which include multi-head self-attention and MLP layers. These blocks reconstruct the missing latent representations using information from the visible regions in each image. From the output of the Transformer blocks, we replace the tokens of non-occluded patches with the original DINOv3 tokens to maintain the quality of the latent representations of the visible parts of the target object and reconstruct the tokens of the occluded parts. Consequently, we obtain the reconstructed image feature tokens $\mathbf{h}_{\text{recon}} \in \mathbb{R}^{N \times L \times d_E}$ corresponding to the $N$ multi-view images by reconstructing masked tokens with our Transformer-based method.

To understand the 3D relationship across multi-view images, the model should be conditioned on camera parameters. We utilize Plücker rays with Adaptive Layer Normalization (AdaLN) (Peebles & Xie, 2023) for camera conditioning at each image feature token, enabling patch-level camera modulation through unique ray representations for each patch. AdaLN with Plücker rays acts as a positional embedding for each image token, facilitating reasoning with camera information within the self-attention mechanisms of the Transformer blocks. Additionally, we utilize PRoPE (Li et al., 2025) for global self-attention layers to inject relative positional information into the model.

### 3.3 Occlusion-aware Feed-forward 3D Reconstruction

With reconstructed image feature tokens $\mathbf{h}_{\text{recon}}$ from the previously described MAE-like reconstruction method, we then feed the tokens into a feed-forward image-to-3D reconstruction model as described in the fig. 2. We utilized pre-trained LaRa (Chen et al., 2024a) as our base multi-view feed-forward 3D reconstruction model due to its efficiency in training and high quality of 3D reconstruction results.

Reconstructed image feature tokens are split for each image as $h_{\text{recon}}^1, \cdots, h_{\text{recon}}^N$, and modulated with adaptive layer normalization with Plücker ray directions. Following LaRa, these features are then lifted to a 3D volume feature by back-projection to a canonical volume with given camera poses and work as an image condition for predicting the Gaussian volume $\mathbf{V}_{\mathcal{G}}$, which represents the completed shape of the target object.

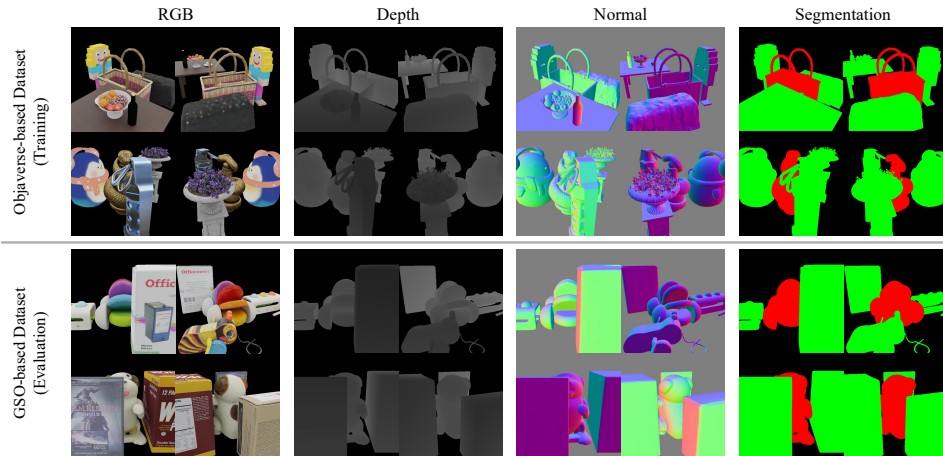

Figure 4: **Example scenes of our cluttered scene dataset.** Training datasets consist of Objaverse objects, and the evaluation dataset is based on the Google Scanned Objects (GSO) dataset.

To account for occlusions in the 3D reconstruction process, we build a *volumetric* obstacle mask and use the mask to obtain information about possibly occluded 3D voxels. Analogous to lifting 2D image features into a 3D space, we lifted the obstacle mask to create a volumetric representation. During this process, a 3D binary occlusion mask was generated for each viewpoint by thresholding the values obtained through grid sampling; values exceeding 0.5 were set to 1, and all others to 0. We then applied group cross-attention between the embedding volume and this binary 3D mask, which enables the model to identify and learn to complete potentially occluded regions.

From the predicted Gaussian volume from our occlusion-aware volumetric Transformer, images of novel views and the mesh of objects are facilitated via the rasterization process and TSDF integration using Open3D (Zhou et al., 2018), following the original process from Gaussian splatting (Kerbl et al., 2023; Huang et al., 2024).

### 3.4 TRAINING OBJECTIVES

For training of our model, we incorporate image reconstruction objectives and regularization terms following Chen et al. (2024a); Zhang et al. (2024). Specifically, we use the MSE loss between the rendered image $\mathcal{I}$ and the non-occluded ground-truth image $\hat{\mathcal{I}}$, calculated only on the non-masked patches. Also, we use the SSIM loss and the Perceptual loss as follows:

$$\mathcal{L} = \mathcal{L}_{\text{MSE}}(\mathcal{I}, \hat{\mathcal{I}}, \tilde{\mathbf{M}}) + \lambda_{\text{SSIM}}\mathcal{L}_{\text{SSIM}}(\mathcal{I}, \hat{\mathcal{I}}) + \lambda_{\text{Perceptual}}\mathcal{L}_{\text{Perceptual}}(\mathcal{I}, \hat{\mathcal{I}}) + \mathcal{L}_{\text{Reg}}. \quad (1)$$

The regularization term $\mathcal{L}_{\text{Reg}}$ consists of distortion and normal regularization components, following Huang et al. (2024); Chen et al. (2024a):

$$\mathcal{L}_{\text{Reg}} = \gamma_{\text{d}} \sum_{i,j} \omega_i \omega_j \, |z_i - z_j| + \gamma_{\text{n}} \sum_i \omega_i \left(1 - \mathbf{n}_i^\top \mathbf{N}\right). \quad (2)$$

The depth distortion regularization term encourages the concentration of the weight distribution, improving geometry reconstruction. The normal consistency regularization ensures that the 2D splats are aligned with the predicted shape's surface.

### 4 DATASET

Existing datasets (Xiang et al., 2018; Kaskman et al., 2019; Tyree et al., 2022) are often limited by a restricted number of object categories or insufficient specificity, which hinders the occluded object reconstruction. To overcome this limitation, we generated a novel training dataset comprising 29,853 occluded object scenes with Objaverse (Deitke et al., 2023) objects. The dataset comprises frames from 955K viewpoints, including RGB, depth, normal, and segmentation maps, rendered using BlenderProc (Denninger et al., 2019). We utilized a filtered subset of objects proposed in LGM (Tang et al., 2025), ensuring a high-quality representation.

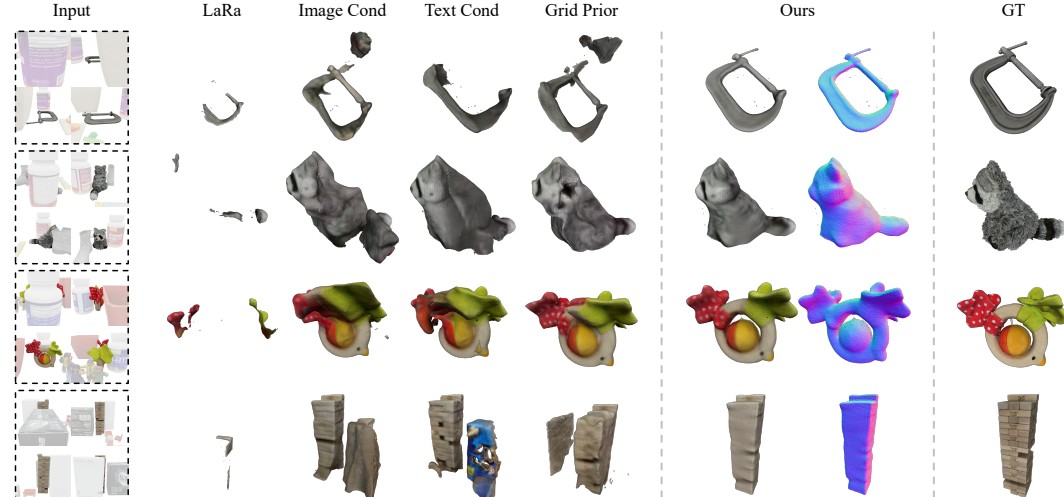

Figure 5: **Qualitative results on our GSO-based evaluation dataset.** To enhance visibility, we emphasize the target object of each scene by brightening the non-target regions in the left images. Additional qualitative results are provided in appendix A. (Please zoom in for more details.)

Each central object was normalized within a bounding box defined by dimensions $[-0.5, 0.5]^3$ in world coordinates and positioned at the origin. To represent occlusion, we placed obstacle objects on the same plane as the central target object and conducted collision tests to ensure that there was no overlap, thereby maximizing the diversity of occlusion. Obstacles were randomly selected from the 30K central objects, creating diverse occlusion patterns. To avoid full occlusion from all viewpoints, obstacles were scaled to half the size of the central object. Each scene includes one central object and three obstacles.

We sampled 32 views from random orientations for each scene, rendered at a resolution of 512×512 pixels. Camera poses were randomly set within a radius ranging from [1.5, 2.2] and an elevation range of [0, 30] degrees.

For evaluation, we employed the Google Scanned Objects (GSO) dataset (Downs et al., 2022), rendering test scenes similarly to the training setup. Each object in the evaluation set was normalized as in training, surrounded by three obstacles. In contrast to the training dataset, the azimuth angles of cameras in the evaluation dataset were uniformly sampled at consistent intervals to provide 360-degree coverage, while the elevation angle was fixed at 20 degrees.

## 5 EXPERIMENTS

### 5.1 EXPERIMENTAL SETUP

**Implementation details.** For the training, we employed the AdamW optimizer with a learning rate of $2 \times 10^{-4}$, scheduled via cosine annealing. The model was trained over 40 epochs on four A6000 GPUs with a batch size of 2, requiring approximately four days to complete. The input and rendering image resolutions were set to $512 \times 512$ pixels, utilizing a DINOv3 image encoder with a ViT-B/16 backbone. Our *Masked Multi-view Volumetric Transformer* architecture consists of six Transformer blocks for 2D image feature reconstruction and twelve blocks for occlusion-aware feed-forward 3D reconstruction.

**Evaluation metrics.** We evaluate the 3D reconstruction quality and 2D novel view synthesis from the completed shapes. To assess the quality of the 3D reconstruction, we report the Chamfer Distance (CD) and F-Score (FS) with a threshold of 0.001. For the 3D metrics, we uniformly sample 100,000 points from the resulting meshes obtained via TSDF-based mesh extraction and the ground truth meshes. We then align both point clouds to the same coordinate system, rescale all meshes to fit within a $[-1, 1]^3$ cube, and perform an Iterative Closest Point (ICP) registration to align the point clouds. For 2D visual quality evaluation, we report the Peak Signal-to-Noise Ratio (PSNR), the

(a) Qualitative results on our real-world captured dataset.

(b) Qualitative comparison between Oct-MAE (Iwase et al., 2024) and our method.

Figure 6: **Additional qualitative comparisons.**

Structural Similarity Index Measure (SSIM), and the Learned Perceptual Image Patch Similarity (LPIPS).

**Baseline methods.** Our baselines are as follows:

- LaRa (Chen et al., 2024a) - LaRa with no inpainting and no additional training for occluded inputs.
- SD (Image Cond) - Inpainting from image and obstacle mask without text prompt conditioning. Based on Stable Diffusion 2 (Rombach et al., 2022).
- SD (Text Cond) - Inpainting conditioned with a text prompt. Based on Stable Diffusion 2.
- Grid Prior - Inpainting with grid prior, proposed by Weber et al. (2024).
- pix2gestalt (Ozguroglu et al., 2024) - 2D amodal completion with image and visible mask.
- OctMAE (Iwase et al., 2024) - OctMAE with a single RGB-D image input sampled from our evaluation dataset.
- Amodal3R (Wu et al., 2025) - Amodal 3D reconstruction model based on 3D object generation model (Xiang et al., 2025).

We utilize our generated obstacle masks for inpainting and use the object name for the text prompt conditions as "*A photo of* {description}". In appendix C, we described detailed information on the text prompt.

**Experimental setup.** For comparison with inpainting methods, we use the same input images as ours for the methods' inputs and employ the inpainting results as the inputs to the LaRa method. Additionally, we developed the qualitative comparison solely to compare OctMAE's result with ours, as OctMAE's problem setting, which utilizes a single RGB-D image of all objects in a cluttered scene, differs from our problem setup.

We utilized our clutter scene dataset, which contains objects from the Google Scanned Objects dataset, comprising 1,030 unseen scenes. We select four novel views for the input of each method for quantitative analysis and use the remaining 12 views for evaluation.

Also, we conduct a zero-shot experiment with real-world captured images to evaluate the in-the-wild performance of our approach. The images are extracted from the frames of real-world captured videos, and we center-cropped the images by adjusting the camera intrinsics. Camera poses and obstacle masks are generated using COLMAP and SAM2 from the center-cropped images.

## 5.2 COMPARISON WITH BASELINE METHODS

As shown in table 1, our method outperforms other generative methods in 3D geometry metrics and 2D novel view synthesis metrics. Since the evaluation dataset consists entirely of unseen data for our model, this result demonstrates that our method achieves zero-shot generalization in reconstructing occluded shapes.

Fig. 5 presents qualitative comparison results on our GSO-based occlusion dataset. Our method exhibits view-consistent completion results in these comparisons through our large dataset and the

Table 1: **Quantitative results on our test dataset based on objects from GSO.** For inpainting-based methods, all inpainted inputs are then reconstructed with LaRa.

| | CD↓ | FS↑ | PSNR↑ | SSIM↑ | LPIPS↓ | Time (s)↓ |
|---|---|---|---|---|---|---|
| LaRa (Occluded Inputs) (Chen et al., 2024a) | 0.263 | 0.313 | 18.916 | 0.903 | 0.135 | **1.707** |
| SD (Image Cond) (Rombach et al., 2022) | 0.187 | 0.454 | 18.461 | 0.875 | 0.171 | 10.61 |
| SD (Text Cond) (Rombach et al., 2022) | 0.182 | 0.454 | 18.739 | 0.875 | 0.169 | 10.49 |
| Grid Prior (Weber et al., 2024) | 0.141 | 0.501 | 22.091 | 0.901 | 0.133 | 24.79 |
| pix2gestalt (Ozguroglu et al., 2024) | 0.163 | 0.508 | 19.606 | 0.888 | 0.146 | 31.63 |
| Amodal3R (Wu et al., 2025) | 0.154 | 0.512 | 17.419 | 0.874 | 0.150 | 9.303 |
| Ours | **0.065** | **0.706** | **25.746** | **0.923** | **0.101** | 2.296 |

*MMVT*. Previous methods, as depicted in fig. 5, generate artifacts or produce multi-view inconsistent inpainting results under our practical obstacle mask setup, leading to inaccurate 3D shape predictions in image-to-3D methods. In contrast, our model accurately predicts occluded shapes by reconstructing the missing latent representations from occluded images.

As demonstrated in fig. 6a, our model reliably reconstructs shapes from occluded regions in real-world images under a zero-shot setup. Our model demonstrates artifact-free novel view synthesis across various scenes, indicating its generalizability derived from learning general information from our large-scale dataset of synthetic objects and its possibility of adapting to real-world tasks.

Additionally, as illustrated in fig. 6b, OctMAE struggles to distinguish individual objects or reconstruct occluded regions when a significant portion of the object is hidden. In contrast, our proposed method effectively reconstructs the geometry and texture of the target object, including its occluded regions, even without utilizing depth data. This suggests our method has potential applications in object separation and completion in real-world cluttered scenes.

## 5.3 ABLATION STUDIES

We conducted an ablation study to evaluate the contributions of our method's key components: 2D image feature reconstruction with a MAE-like architecture, and 3D occlusion-aware reconstruction via

Table 2: **Ablation study on the key components of our *MMVT*.**

| | CD↓ | FS↑ | PSNR↑ | SSIM↑ | LPIPS↓ |
|---|---|---|---|---|---|
| LaRa (Occluded Inputs) | 0.263 | 0.313 | 18.916 | 0.903 | 0.135 |
| + 2D Reconstruction | 0.098 | 0.587 | 23.671 | 0.914 | 0.111 |
| + Volumetric Mask | 0.076 | 0.655 | 24.810 | 0.919 | 0.108 |

cross-attention with volumetric obstacle masks. In the ablation study, due to resource limitations, we use model variants that were trained for 20 epochs. To analyze the impact of our 2D image feature reconstruction with a large-scale occlusion dataset, we compared the original LaRa implementation with our model.

In table 2, *2D Reconstruction* refers to 2D image feature reconstruction via a multi-view MAE-like architecture; *Volumetric Mask* indicates the occlusion-aware 3D reconstruction with cross-attention between volumetric latents and 3D-lifted obstacle masks. From the table,

## 6 CONCLUSION

We presented the data-driven approach for multi-view consistent reconstruction of missing image latent representations from occlusions. Our method employs a data-driven approach, utilizing large datasets of cluttered scenes. It leverages multi-view consistent reconstruction, global reasoning over multiple latent and mask tokens, and occlusion-aware 3D reconstruction via volumetric masks. Through our novel multi-view consistent masked area reconstruction and extensive dataset, we achieve high-quality occluded area completion with efficient inference. Specifically, our model outperforms previous methods and requires only five seconds for inference. In future work, we plan to extend our method to real-world tasks such as multi-object manipulation in robotics. The limitations are detailed in appendix E.

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
