# OpenReview forum: "Occluded 3D Object Reconstruction via Masked Multi-view Volumetric Transformer"
_ICLR.cc/2026/Conference — ICLR 2026 Conference Withdrawn Submission_

### Official Review · Reviewer_W1bq · 2025-10-26

**Soundness:** 3
**Presentation:** 3
**Contribution:** 3
**Rating:** 6
**Confidence:** 5

**Summary:**

## Summary

This paper addresses the problem of multi-view amodal 3D reconstruction, which takes multi-view images with camera poses as input and outputs parameters of 2D Gaussians—including RGB, depth, and alpha maps. The proposed pipeline builds upon LaRa, first constructing 3D feature volumes by lifting 2D DINO features into a canonical volume, and then applying a volumetric transformer to reconstruct a Gaussian volume. The key contribution lies in handling incomplete input images: the authors introduce a MAE-inspired module for completing DINO features and integrating both the completed volumes and volumetric occlusion masks into the 3D generation process.

**Strengths:**

## Strengths

1. This paper presents a novel pipeline for multi-view amodal completion, formulating the problem of the generation of Gaussian volumes and designing a MAE-like latent reconstruction method for feature completion.
2. The authors also contribute a new large-scale training dataset comprising 29,853 occluded object scenes built from Objaverse, which provides a valuable foundation for future research.
3. The quantitative results in Table 1 are highly promising and demonstrate the effectiveness of the proposed approach.

**Weaknesses:**

## Weaknesses
1. Fairness of Comparisons: To the best of my knowledge, methods such as Amodal3R and other image-based occluded object generation approaches typically operate on unposed images. Therefore, a direct comparison with these methods may not be entirely fair.

2. Role of the DINO-MAE Module: Although the authors include ablation studies with and without the 2D reconstruction module, it remains unclear whether the model could directly infer 3D Gaussian volumes from incomplete volume features and volumetric masks alone—an approach that has been validated in works like Amodal3R. Further analysis on this point would help clarify the necessity and contribution of the 2D reconstruction component.

**Questions:**

The author should provides a more fair comparisons with the baselines. I am open to adjust the rating.

---

### Official Review · Reviewer_4j3T · 2025-10-30

**Soundness:** 3
**Presentation:** 3
**Contribution:** 3
**Rating:** 6
**Confidence:** 4

**Summary:**

This paper presents a novel method for reconstructing 3D objects from multi-view images under occlusion. The proposed approach, *Masked Multi-view Volumetric Transformer (MMVT)*, leverages a masked autoencoder-inspired architecture to reconstruct missing latent representations from occluded regions, ensuring multi-view consistency. The authors also contribute a large-scale synthetic dataset of cluttered scenes for training and evaluation. Experimental results demonstrate that the method outperforms existing baselines in both quantitative metrics and inference speed, completing 3D reconstructions in approximately five seconds.

**Strengths:**

1. The proposed MMVT model effectively handles occluded object reconstruction by integrating multi-view reasoning and volumetric obstacle masks, achieving state-of-the-art performance in both 3D reconstruction and novel view synthesis.
2. The method is more efficient, with inference times faster than most competing approaches (e.g., inpainting-based or diffusion-based methods).
3. The paper is overall well-written and should be easy to follow.

**Weaknesses:**

1. While quantitative comparisons with methods like pix2gestalt and Amodal3R are provided, the lack of corresponding qualitative visualizations makes it difficult to fully assess the visual superiority of the proposed method. Including more visual comparisons, especially rotating videos of reconstructed objects, would strengthen the results.
2. The real-world qualitative results in Fig. 6 appear less convincing, raising concerns about the model’s generalization to real-world scenes. Results on established real-world datasets such as ScanNet++ or Mip-NeRF 360 would help validate its practical applicability.

**Questions:**

In the paper, the obstacle mask is defined as the union of all instance masks excluding the target object. However, this mask may include regions that do not actually occlude the target. Would it be more straightforward and effective to treat the entire image area outside the target object mask as the obstacle mask? This could simplify the input conditions.

---

### Official Review · Reviewer_vprQ · 2025-10-31

**Soundness:** 2
**Presentation:** 1
**Contribution:** 2
**Rating:** 2
**Confidence:** 3

**Summary:**

Authors propose to solve a problem of reconstructing occluded 3D shapes using a novel masked multi-view volumetric transformer method which reconstructs missing image features token at masked patches in a feed-forward manner.

Specifically, the authors use DINOv3 image features + plucker ray modulation. Feedforward reconstruction has MAE-like architecture with global + frame-level self-attention over concatenated multi-view tokens.

**Strengths:**

- Author's approach of patching final image tokens with unmasked tokens is, to my knowledge, novel
- This method of getting image feature token, in combination of using 3D obstacle volume mask with group cross-attention is novel, to my knowledge
- According to quantitative evaluation, the method appears to be producing SOTA results on author's test dataset

**Weaknesses:**

1. The main constribution appears to be in handling occluded images and incorporating occlusion masks in generating image tokens, while existing approach (pre-trained LaRa) is used afterwards for feed-forward reconstruction. I'm doubtful of the value of this contribution alone, yet willing to reconsider if other reviewers think otherwise.
2. The choice of patching image tokens with unmasked tokens is not ablated.
3. Albeit strong quantitative results (Table 1), visual presentation is very unconvincing (e.g. Fig 1, Fig 6) and is clearly behind state-of-the-art in 3D reconstruction
4. The custom dataset that authors produce cannot be considered a solid contribution in it's current form. Additionally, authors make no promises to release the dataset for community
5. Authors select weak method (LaRa) for testing their inpainting methods

**Questions:**

1. Please clarify what are the main contributions of the paper, apart from incorporating occlusion masks while getting image tokens for feed-forward reconstruction.
2. Please clarify why LaRa was selected as 3D reconstruction pipeline with inpainting methods? I suggest using one of the most recent methods (Trellis / Step1X-3D / Hunyuan3D)
3. Please provide more qualitative evaluations supporting statements in Table 1. Specifically, visual comparison b/w your method, inpainting methods with stronger baseline (see 1.), and Amodal3R. For Amodal3R specifically, the visuals in the paper are significantly stronger than the work in question, so I'd appreciate to hear more details to explain this discrepancy.
4. Please provide ablation on patching unmasked image tokens (Fig. 2), it's not clear how helpful this decision is.
5. L70 - which artifacts? Could you provide examples / elaborate?
2. Please clarify whether you plan to release the code and dataset.

---

### Official Review · Reviewer_e5s5 · 2025-11-02

**Soundness:** 2
**Presentation:** 1
**Contribution:** 2
**Rating:** 4
**Confidence:** 4

**Summary:**

This paper proposes a transformer-based model for reconstructing partially occluded 3D shape. Goal is fast feedforward inference with multi-view consistency. It consists of two parts: a feature extractor and a shape decoder. The feature extractor fuses different views via attention, and each view is processed with the frozen DINOv3 model. Since instance segmentation is required as input, we know which patch/token is an occlude that we are not reconstructing, so the obstacle masks can be lifted into 3D just like the image features. The shape decoder then produces 3D shape from these feature and obstacle mask volumes.  Shape representation is gaussian splats.

The feature extractor is built on top of a MAE but made multi-view. View information is injected as pluecker ray directions via adaptive layer norm. The decoder is a pre-trained LaRa. The loss is standard re-rendering loss along with SSIM and perceptual metrics. Regularization is depth distortion and normal consistency.

**Strengths:**

Feedforward, multi-view-consistent 3D reconstruction is a very practical and important problem to solve. The paper tackles exactly this. Good topic.

The quality achieved on non-standard shape looks good, and quantitative results show that the model is strong and fast.

**Weaknesses:**

The occlusion setup feels quite artificial to me – the procedurally generated scenes are nowhere close to what occlusions look like in real life. I understand how training can be hard without ground truth on real occluded scenes, but at least the trained model should be put to test on such scenes. However, the current test cases are similar-looking synthetic scenes (objects being scanned and real is not helping much here).

Similarly, requiring instance seg masks is another major drawback, limiting the practicality. Either the model can work with a quick scribble defining the object of interest, or we should be after a non-object centric model, if we are for practical approaches that can actually be useful.

Presentation needs more work. Multiple places with broken contents or grammar. Examples: broken caption in L170-173. Maybe due to this, I found this paper hard to follow.

**Questions:**

What do the results on real scenes with occlusion look like?

Can the model work with a quick scribble instead of pixel-aligned ins seg masks?

Does the ability to complete occluded parts come *more* from geometry priors learned from data or multi-view attention design? A or B? Why?

---

### Note · Authors · 2025-11-12

**Comment:**

Thank you for your reviews. Based on the comments, we have decided to withdraw our submission. We truly appreciate the constructive suggestions, and we will incorporate them to improve the work for future publication.

**Withdrawal Confirmation:**

I have read and agree with the venue's withdrawal policy on behalf of myself and my co-authors.